# Impact of being an adolescent mother on subsequent maternal health, parenting, and child development in Kenyan low-income and high adversity informal settlement context

Manasi Kumar[1]*, Keng Yen Huang[2]

1 Department of Psychiatry, University of Nairobi, Research Fellow, University College London, London, United Kingdom, 2 Department of Population Health, New York University School of Medicine, New York, NY, United States of America

* m.kumar@ucl.ac.uk

**Data Availability Statement:** All relevant data are within the paper and its Supporting Information files.

## Abstract

### Background

Women who have experienced adolescent pregnancy and early motherhood are disproportionately affected in terms of their health and parenting capabilities, as well as their offspring's health. Guided by Stress Process and Social Determinants of Health (SDH) Frameworks, which posit that multiple sources of stressors and structural determinants of adolescent pregnancy influence adolescent mothers' subsequent health and quality of parenting (Xavier et al 2018, McLoyd 1998, Conger et al 2010, Gipson et al 2008). These dynamics then further impact offspring's outcomes. Using an Integrated Stress-SDH Process for Health Disparities model and we test on whether early motherhood is associated with and subsequent maternal and child health from two informal settlements in Nairobi.

### Methods

A cross-sectional design with 394 mothers of 2–16 years old children who sought maternal and child health services at Kariobangi and Kangemi public health centers between October 2015 to April 2016 were recruited. Participating mothers were asked questions related to their adolescent pregnancy history, their current health, wellbeing, and parenting practices, and their child's health. Structural equation modeling (SEM) was utilized to examine hypothesized mediational pathways that adolescent pregnancy history has negative influences on women's health and parenting during adulthood, which also influence their child's health and development.

### Results

Our study supports that women with a history of adolescent motherhood have poor physical and mental health outcomes as adults after adjusting for demographic confounders. SEM results partially support the Stress-SDH Process model that history of adolescent pregnancy had negative consequences on women's adulthood health, which also negatively impacted offspring's physical and mental health.

**Funding:** Fogarty International Center/NIH K43 TW010716 award to Manasi Kumar from 2018-2023 National Institutes of Mental Health U19 MH110001-01- award to Manasi Kumar and Keng Yen Huang.

**Competing interests:** The authors have declared that no competing interests exist.

## Conclusion

Consistent with the Stress Process and SDH literature, we found consistent cross-cultural literature that adolescent pregnancy set the stage for, subsequent poor maternal health and child outcomes. Although history of adolescent pregnancy and motherhood was not necessarily associated with negative parenting, consistent with parenting literature, negative parenting was associated with poor child mental health. Findings suggest importance of providing integrated care that address health and parenting needs to optimize offspring's development in instances of early motherhood.

## Background

Adolescent pregnancy and early motherhood continue to be a global public health burden. In Sub-Saharan Africa, adolescent girls continue to experience disproportionately high burden of sexual and reproductive ill health, including early pregnancy and mental health problems [1]. In 2014, the Kenya demographic health survey reported that Teen pregnancy and motherhood rates in Kenya stood at 18%. About 1 in every 5 adolescent girls had either had a live birth, or was pregnant with her first child. Rates increase rapidly with age: from 3% among girls at 15 years old, to 40% among girls at 19 years old [2]. However, between July 2016 and June 2017, the Kenyan Ministry of Health recorded almost 350,000 pregnancies among young women aged between 15 and 19 years [3]. In 2019 the Global Childhood statistics report that Kenya had the third-highest teen pregnancy rates globally with 82 births per 1,000 births [4]. These figures suggest a significant rise in adolescent pregnancies in Kenya. Adolescent pregnancy can have significant negative consequences on adolescent girls' short- and long-term health outcomes as mothers, as well as their offspring's health. Despite this broad finding, there are limited studies available. For example, age-period effects on women's health remains relatively unexplored in public mental health. It remains unclear whether adolescent motherhood will have accumulated negative effect as women age, such that chronic strain will lead to deterioration of health and worsening parenting and child outcomes for older age group of women than younger age of women in low-and-middle income countries (LMICs). In addition, limited studies have investigated whether stress process model is supported in LMICs.

### History of adolescent pregnancy and early motherhood and implications for maternal health: Mapping health and psychosocial impacts on mothers and offspring

Challenges associated with early and unintended motherhood are enormous. Two theoretical frameworks can explain the mechanisms and consequences of unintended adolescent pregnancy. According to the *Family Stress Model*, level of adolescent pregnancy and environmental-related stress can influence quality of parenting, which impacts child outcomes [5, 6]. The *Social Determinants of Health* (SDH) posits that social and structural environment and available resources for adolescent mothers influence their and their offspring's health [7]. Both frameworks recognize complex interplay and stress experiences of the social and economic systems that people (adolescent mothers) live in and their impacts on health behaviors and health outcomes of family members. For adolescent mothers in LMICs and in Sub-Saharan Africa, unintended motherhood can lead to experiences of stigma, poverty, low education, unemployment, low support, and food insecurity [8]. The experience of these multiple

stressors, including adverse environment and structural inequality, can lead to girls' development of poor physical and mental health, and regressive sexual and reproductive and behavioral practices (e.g., inadequate parenting to support offspring). The combination of stress and adverse social determinants will also lead to disparities in offspring's health and development outcomes [9, 10].

A body of research from developed countries has tested different theoretical frameworks and documented that adolescent pregnancy has long–term adverse effects for both mothers and children [11]. Adolescent pregnancy has been found to be associated with increased risk for maternal physical and mental health problems [12–15]. It also contributes to subsequent lower socioeconomic status, poor physical health, and higher mortality in adulthood [11]. Furthermore, adolescent pregnancy increases the likelihood for severe medical and obstetric complications. One study from UK found that the early age at first birth is associated with poorer mental health among women in their fifties [16]. Another study with over 4000 women participants from Australia found that young mothers, particularly teenage mothers, were at high risk of poor mental health than mothers aged 25 years and above. There was also an indication that health disparities between teenage and normal birth age mothers emerged over time, with increased risk for poor health for adolescent mothers as they aged [17]. However, such research in LMIC remains in its infancy.

## Adolescent motherhood and implications for maternal and offspring health in Sub-Saharan African context

Research from developed countries has also documented negative consequences of adolescent motherhood on offspring's health and development. Specifically, children of adolescent mothers have been shown to be more likely to have delays in cognitive and language abilities, including poor academic achievement and mental health outcomes [18–20]. In SSA contexts, research on mental health consequences of adolescent motherhood is still lacking. It is likely that the impacts of adolescent pregnancy on motherhood, and maternal and child wellbeing would be greater because of the likelihood of experiencing multiple adversities before and during the motherhood transition.

From a developmental perspective, because adolescents are not fully individuated from their own families, developmentally mature in emotion and cognitive ability, and not having adequate financial resources, it is challenging for them to raise their offspring in an optimal way [21]. In addition, in a collectivist culture of SSA, motherhood is generally viewed with great positive energy and women are ranked socially on the basis of their maternal status. However, adolescent mothers in SSA tend to be *single or in conflictual marriages*, *being out of school*, *having negative peer models*, *and often belong to families with poor functioning* and instability (i.e. unstable family structure, absence of a father figure, and low support) [21]. For adolescents growing up in informal settlements, they also face *additional challenges*, *including exposure to high violence, maltreatment experience during childhood, with an absence of male partners or family support, and engaging in risky early and unprotected sexual exposure* [22, 23]. Moreover, adolescent mothers in SSA tend to experience greater *barriers and stigma in navigating health services*, being taken seriously by the health workers and relatives, which further exacerbates their stress and negative parenthood experience [24–27]. These cumulative stressors experienced during adolescent motherhood aggravates parenting stress [28] and increase the risk for depression [29].

In summary, the experience of multiple adversities, stress, and anxiety during pregnancy in adolescence and throughout the motherhood process could diminish the ability of the young mother to be *self-efficacious* [23, 30], conflictual marriages and exposure to gender-based

violence, lack of empowerment and opportunity to make decisions in the family can lead to as much disadvantage in child welfare as disrupted unions in case of adolescent mothers. Single motherhood has been associated with poorer physical and mental health, and higher risk of mortality for children globally [31]. All these adversity experiences can increase the risk for poor maternal and child physical and mental health [32].

## Objectives and associated research hypotheses

Given this background, our extended the limited adolescent motherhood literature in SSA, and examined three specific research questions

1. Guided by the Family Stress and SDH Frameworks to examine whether the history of adolescent pregnancy is associated with poor adulthood health and parenting (after controlling for demographic and social determinant factors) [12, 33, 34]. We would also examine whether this association is differed by age-period (studying whether older age of women with a history of adolescent motherhood would have the worse health and parenting outcomes than younger age of women with or without a history of adolescent motherhood because of chronic strain and accumulated stressor/SDH and negative experiences). *We have defined the history of adolescent pregnancy (or adolescent mothers) as mothers who had their first child at ages 18 or younger.*
   *Hypothesis*: *Mothers who have children during adolescence are likely to have more health and parenting problems than mothers who had children as adults. Also, adolescent motherhood history is associated with deterioration of health and parenting outcomes as women aged (age-period effect).*

2. To examine whether the history of adolescent pregnancy contributes to health disparities in offspring (as shown to have more physical and mental health problems in offspring of adolescent mother group).
   *Hypothesis*: *Mothers who have children during adolescence are more likely to have offspring with more physical and mental health problems than mothers who had children as adults.*

3. To study the Stress-SDH Process mechanism (a path model) by examining whether adolescent pregnancy history has long term impacts on girls' adulthood women health and parenting, which would further impact their offspring health (Fig 2).
   *Hypothesis: Adolescent motherhood would be associated with poor maternal health and negative parenting practices, which would further associate with poorer child outcomes.*

## Methods

### Participants

The study utilizes a cross-sectional study design. Mothers of 2–16 years old children and young people seeking maternal and child health services at Kariobangi and Kangemi public health centers were invited to participate. A mother in this study was defined as biological birth parent. A sample of 394 mothers who over 18 years of age were recruited. Mothers who were psychologically unstable or physically unable to participate were ineligible for the study. Table 1 summarizes the demographic characteristics of the study sample. Our study sample included 394 mothers, with 126 mothers (32%) having their first child at age 18 or below (the adolescent mother group), and 268 mothers (68%) having their first child after age 18 (the non-adolescent mother group). Over 97% of the sample followed different denominations of Christianity and 61.7% reported experiencing food insecurity. About 67% of the mothers were married or live with partner.

**Table 1. Family demographic by adolescent motherhood status.**

| | Total (n = 394) | Adolescent mothers (n = 126) | Non-adolescent mothers (n = 268) | OR | p |
|---|---|---|---|---|---|
| *Demographic Characteristics* | Mean (SD) or % | Mean (SD) or % | Mean (SD) or % | | |
| Education- | | | | .43 | < .001 |
| Primary or less (ref) | 69.8% | 81.0% | 64.6% | | |
| Secondary or higher | 30.2% | 19.0% | 35.4% | | |
| Marital Status | | | | .75 | .213 |
| Single /divorced/widowed (ref) | 33.0% | 31.0% | 37.3% | | |
| Married or live with partner | 67.0% | 69.0% | 62.7% | | |
| Employed | | | | .69 | .087 |
| Yes | 51.5% | 45.2% | 54.5% | | |
| No (ref) | 48.5% | 54.8% | 45.5% | | |
| Food Security (ref) | 38.3% | 34.7% | 41.4% | 1.52 | .066 |
| Insecurity | 61.7% | 68.3% | 58.6% | | |
| Religion | | | | .37 | .128 |
| Other (ref) | 2.3% | 4.0% | 1.5% | | |
| Christian | 97.7% | 96.0% | 98.5% | | |
| Number of Household Members | 4.82 (1.57) | 5.21 (1.64) | 4.64 (1.50) | - | .002 |
| Number of Children <18 years old | 2.84 (1.35) | 3.15 (1.41) | 2.69 (1.29) | - | .001 |
| Parental Age | 31.62 (7.01) | 29.95 (7.11) | 32.40 (6.82) | - | .001 |
| Social Support (4) | 3.76 (1.03) | 3.64 (1.12) | 3.82 (.97) | - | .102 |
| Child Gender-Boy | 47.2% | 50.4% | 45.6% | 1.21 | .376 |
| Child Age | 6.89 (3.05) | 7.66 (2.81) | 6.53 (3.10) | - | .001 |

*Note*: Percentages (%) showed in the 1st column are demographic characteristics for the entire sample. OR (= Odds ratio) and *p* values are a result of the unadjusted association between adolescent mother and non-adolescent mothers (1 = yes, 0 = no). (ref) = reference group.

## Setting and data collection procedure

Participants were recruited between Oct 2015 to April 2016 from two maternal and child health (MCH) centers located in Kariobangi and Kangemi informal settlement communities of Nairobi. Eligible women were approached to obtain oral or written informed consent. The consent information was shared verbally, and a written consent form (with written information about the study) was given at the time of obtaining consent. A signed consent form was documented for parents who agreed to participate. Consent procedure and interviews were carried out in Kiswahili and English, depending on the preference of the participants. The study procedures and method of consent were approved by the Institutional Review Board of the Kenyatta National Hospital/University of Nairobi (IRB number: KNH/UoN-ERC Ref. P520/08/2015).

Participating parents were asked questions related to their childhood experience, their current health and wellbeing, parenting practices, and their child's health. Only one child was targeted for families with more than one child (age range 2–16 years). Children did not directly participate in the study activities, and they were not the subjects of this study. Participants who identified with a high depression score (cut-off of 10 or more based on our interview questionnaire Kessler Psychological Distress Scale) or requested for other mental health services (due to stress or anxiety) during the data collection interview were referred to the facility mental health clinic or provided with some onsite psychosocial support.

## Measures

**Maternal physical and mental health.** Three self-report measures were used to assess maternal physical and mental health. The *Kessler Psychological Distress* Scale [35] (K10; 10

items, α = .94 based on the current study sample) assessed anxiety and depressive symptoms that mothers experienced. Mothers rated 10 symptom items on a 5-point scale (1 = none of the time; 5 = all of the time). A total score was created for 10 symptom items. Based on the recommendation, a clinical cut-off score of 25 was also used to estimate the prevalence for mental disorder. Women with a score of 25 or above would suggest a high likelihood of developing a mental disorder. The scale has been demonstrated with good predictive validity using Structured Clinical Interview for DSM-IV (SCID). *Maternal health* was assessed based on mother perception of overall health and quality of life (2 items; α = .70 for the study sample) on a 5-point scale (1 = poor, 2 = fair, 3 = good, 4 = very good, 5 = excellent) [36]. A mean score ≤ 2 indicates poor health. *Adolescent pregnancy status* (1 = give birth to the first child at age 18 or younger, 0 = give birth to the first child ≥ 18 years) was also used to estimate adolescent risky health behavior. Inter-correlation (*r*) between maternal mental health (K10) and maternal health was 0.34, and between adolescent pregnancy and mental health were 0.16 (all with *ps* < .05), suggesting sound validity of the measures for our sample.

**Parenting.**   Three areas of parenting were assessed. Constructs that had been identified as significant predictors for child development in the literature were targeted. *Parental emotion socialization* practices were assessed using the Responses to Children and Emotion Questionnaire (RTCE) [37]. Parents were asked to report the frequency with which his/her child experienced certain emotions over the past month (i.e., sad/down, angry/frustrated, fearful/anxious); then the parent rated how often he/she used different socialization strategies (i.e., reward, punish, neglect) in response to child emotions on a 5-point Likert scale (1 = Never; 5 = Very Often). For instance, the parent is asked the frequency with which they used the following strategy over the past month: "When my child was sad, I comforted her/him." For this study, the summary score of Discouragement of Emotional Socialization (15 items, combine punish and neglect of sadness, anger, and fearful, α = .92) was used. *Parental harsh discipline practice* was assessed using the Parenting Practices Interview (PPI) [38] (6 items; α = .77 for the study sample). Parents were asked to report the frequency they used yell, threaten, hit or spanking practices about an average week in the past month when their child misbehaves on a 4-point scale (0 = never, 3 = almost always). *Conflicted Parent–Child Relationship* was assessed using the Parent–Child Relationship Scale [39] (9 items; α = .62 for the study sample). Parents were asked to rate how much they agree or disagree with the relationship statements, such as "my child and I always seem to be struggling with each other" that they have with the child on a 5-point scale (1 = strongly disagree, 5 = strongly agree). Three study measures have been used with diverse ethnic populations in studies conducted in high-income countries, and have also been validated with our studies in LMICs (i.e., Uganda, Ghana) [36, 40].

**Child physical and mental health.**   *Child Physical Health* was measured with two global items. Parents were asked to rate their overall perception of their child's health on a 4-point scale (1 = poor health; 4 = very good or excellent health) and their perception about their child's tendency to get physical illness on a 3-point scale (1 = certainly true, 3 = not true). Both items are statistically significantly related (r = .46; α = .63); therefore, we created a sum score. A higher score indicates good health [36]. Child Behavior Check List (Achenbach, 1991) was used to assess child internalizing and externalizing problems on a 3-point scale (0 = not true, 2 = very or often true). Internalizing disorders present with an inward directed emotional and behavioral distress such as depression and externalizing disorders present with a propensity towards expressing distress outwards such as conduct disorder. Two CBCL age versions and 3 norm samples were used to score standardized T scores (1.5–5 version, 6–18 version for 6–11 years old and for 11–18 years old). The research team carried out a formal translation of the 6–18 years with permission from the CBCL author Thomas Achenbach. Scale reliability was adequate for both internalizing problem (α = .76-.85 based on our

Kenyan 3 age-group samples) and externalizing problem (α = .87-.89 based on the current Kenyan samples).

**Demographic and covariates.** Household size, number of children in home, marital status (1 = married/live with a partner), maternal education (1 = secondary or higher), employment status (1 = yes), child sex (1 = male, 0 = female), and maternal and child age (continuous variable) were assessed. Food insecurity (using Household Hunger Scale; 3 items; if any item was rated as yes was considered as food insecure [41] was also included as a family financial resource indicator.

## Data analysis

To examine group differences between mothers with and without history of adolescent motherhood on their current adulthood health, parenting, and child health outcomes (Objectives 1 and 2), a series of adjusted linear and logistic regression analyses were conducted. Analyses were conducted separately for each outcome and adjusted for 8 potential confounders, including maternal age, household size, number of children in the home, marital status, child age, child sex, and education and employment status. To test the Stress Process/SDH model, we examine a path model. We study whether adolescent motherhood is associated with poor adulthood health and parenting, which further associated with poor child health outcomes. The path model was tested using Structural Equation Modeling (SEM), allowing for (a) path analysis of the model in its entirety including mediating mechanisms, (b) correlations between all model variables within each domain (parenting contextual factors and child outcomes); and (c) adjustment for potential confounders (i.e., parental age, education, number of children at home, household size, child age, and food insecurity). Mplus 7 was used for SEM, and the maximum likelihood estimation method (ML) was applied. The fit of the SEM models were evaluated using Chi-square ($\chi$2 > .05 or $\chi$2/df ratio less than 3.0), root mean square error of approximation (RMSEA < .08), and comparative fit index (CFI > .95) [42].

## Results

### Comparisons between mothers with and without history of adolescent pregnancy

Table 1 summarizes characteristic differences (socio-demographics and SDH) between mothers with and without a history of adolescent pregnancy. Table 2 summarizes group differences on maternal health and parenting and offspring's health outcome differences.

**Demographic and SDH characteristic comparisons between two groups.** As shown in Table 1, adolescent and non-adolescent mothers were significantly different on *age* and *household composition*. Specifically, the adolescent mother group was younger and tended to have higher number of family members and children in the household than the non-adolescent mother group. The mean age for the adolescent mothers' group was 29.9 (SD = 7) years, and the comparison group was 32.4 (SD = 6.8) years. The number of family members was higher in the adolescent mother group (5.21 (SD = 1.64) family member and 3.15 (SD = 1.41) children under 18 years of age) than the comparison group (and 4.64 (SD = 1.50) family member and 2.69 (SD = 1.29) children). We also found a significant difference in the education determinant between groups. Mothers with adolescent pregnant history were less likely to have secondary or higher education (19%) in comparison to mothers without a history of adolescent pregnancy (35%). There were also trend findings showing that mothers with adolescent pregnant history were less likely to be employed (45% vs. 55%) and more likely to experience food insecurity (68% vs. 59%) than mothers without a history of adolescent pregnancy.

**Table 2. Parental health, parenting, and child functioning by adolescent motherhood status.**

| | Total (n = 394) | Adolescent mothers (n = 126) | Non-adolescent mothers (n = 268) | B (SE) or OR ⓡ | p |
|---|---|---|---|---|---|
| | Mean (SD) or % | Mean (SD) or % | Mean (SD) or % | | |
| *Maternal health* | | | | | |
| Overall Physical Health | 2.45 (.82) | 2.35 (.82) | 2.49 (.81) | -.19 (.09) | .053 |
| Mental Health (K10) | 22.11 (8.33) | 23.89 (8.48) | 21.28 (8.14) | 2.07 (.93) | *.026* |
| At-risk for mental disorder (K10≥25) | 31.5% | 43.7% | 25.7% | 2.13 ⓡ | *.005* |
| *Parenting* | | | | | |
| ES-Discourage negative emotion | 1.53 (.74) | 1.67 (.89) | 1.46 (.66) | .23 (.09) | *.011* |
| PPI-Harsh-Discipline | 1.32 (.72) | 1.35 (.75) | 1.30 (.71) | .05 (.09) | .546 |
| Conflicted Parent-Child Relationship | 2.30 (.59) | 2.27 (.67) | 2.31 (.56) | -.01 (.07) | .874 |
| *Child Health* | | | | | |
| Child Overall Health | 5.39 (1.02) | 5.26 (1.14) | 5.44 (0.95) | -.13 (.12) | .301 |
| CBCL-Externalizing | 46.52 (10.53) | 47.09 (11.03) | 46.25 (10.30) | .35 (1.23) | .774 |
| CBCL-Internalizing | 49.11 (10.44) | 50.94 (10.81) | 48.25 (10.17) | 2.87 (1.23) | *.020* |

*Note*: OR (= Odds ratio) and *p* values are results from adjusted associations using liner regression (for continuous predictors) or OR (for categorical K10 predictor, 1 = at-risk for mental health problem, 0 = not at-risk). All analysis adjusted for potential confounders, including maternal age, household size, number of children in home, marital status (1 = married/live with partner), education (1 = Secondary or higher), employment status (1 = yes), child age, and child sex (1 = male).

**Adulthood health and parenting comparisons between two groups.** On *maternal health*, findings from the adjusted regression analysis (see Table 2) showed that adolescent mothers had significantly poorer adulthood physical health than non-adolescent mothers. In addition, the odds for developing mental disorder in adulthood was about twice higher for those who became mothers during their adolescence than those who become mothers after age 18. About 43.7% of mothers in the adolescent-mothers group were identified to be at-risk for mental disorder compared to 25.7% mothers in the non-adolescent mother group.

On *parenting*, we found that adolescent mothers were more likely to use negative parenting practices than non-adolescent mothers, especially in emotional socialization practices. Adolescent mothers were more likely to discourage their child's emotional expression by punishing or neglect their child's negative emotions than non-adolescent mothers ((M (SD) = 1.67 (.89) vs. M(SD) = 1.46 (.66)). However, inconsistent with our hypothesis, we did not find group differences in mothers' harsh discipline practices and the quality of parent-child relationship after adjusting for confounders.

**Age-period effect.** To understand whether there is an age-period difference for the impacts of adolescent pregnancy on adulthood health and parenting (or whether adolescent motherhood history is associated with deterioration of health and parenting outcomes as women aged) due to cumulated negative experiences, we examined maternal current health status/patterns stratified by age groups and adolescent motherhood status. ANCOVA analysis was conducted by including adolescent motherhood status, age groups (≤ 25, 26–30, 31–35, and ≥ 36 years-old age groups), and interaction of age-by-adolescent motherhood status as the predictors, with adjustment of confounders). For *maternal physical health*, overall, we found adolescent mothers had less optimal physical health than the non-adolescent mothers across all age groups ($F$ (1, 376) = 4.01, p = .046). The patterns did not differ by age group (i.e., with no significant interaction of age-by-adolescent motherhood group). Our findings suggest a history of adolescent pregnancy has a stable life-long negative impact on maternal physical health (see Fig 1(A)). For *maternal mental health*, a somewhat different pattern emerged. ANCOVA showed significant adolescent motherhood effect ($F$ (1, 376) = 8.38, p = .004), and

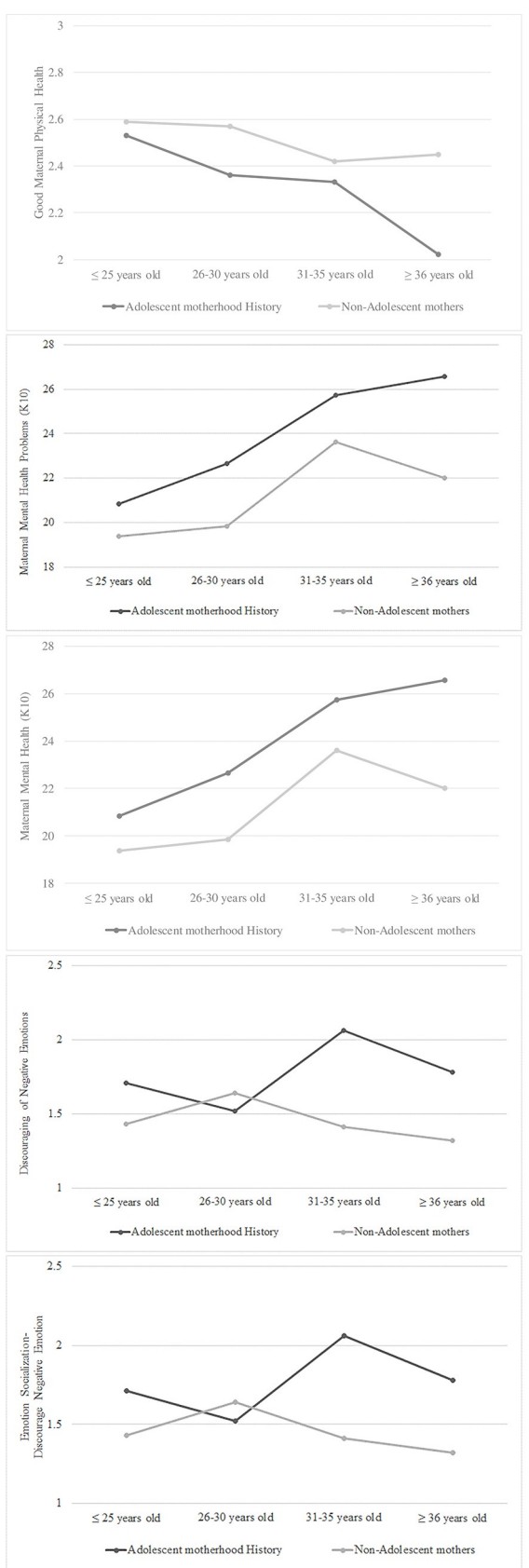

**Fig 1. Maternal health by adolescent motherhood history status and age group.** (a) Maternal Physical Health by Adolescent Motherhood History Status and Age Group. (b) Maternal Mental Health by Adolescent Motherhood History Status and Age Group. (c) Parenting-Discourage Negative Emotion by Adolescent Motherhood History Status and Age Group. ***Note.*** Mean values presented were adjusted means based on univariate ANCOVA analyses, adjusting for potential confounders household size, number of children in home, marital status (1 = married/live with partner), education (1 = Secondary or higher) and employment status (1 = yes). N for ≤ 25, 26–30, 31–35, and ≥ 36 year-old age groups were 28, 56, 21, and 20 for the adolescent mother group, and 45, 7, 73, and 74 for the non-adolescent mother group.

age group effect ($F$ (3, 376) = 5.16, p = .002), but no significant age-by-adolescent motherhood group interaction effect. As shown in Fig 2(B), mothers with a history of adolescent pregnancy tended to have higher mental health problem across life span than mothers who did not have history. Also, across age groups, Kenyan mothers' mental health tends to deteriorate as they got older, and the patterns were consistent for mothers with and without adolescent pregnancy history. Findings indicate a need for mental health preventive intervention for young adult women in Kenyan. For *parenting*, we found a significant age-by-adolescent motherhood group interaction effect on one parenting—Discouraging Negative Emotion in socializing with the child ($F$ (3, 377) = 4.58, p = .004). We found women with adolescent motherhood history have significantly worse parenting (more discourage negative emotion) in the older age group than the younger age group (see Fig 1(C)). We did not find a significant age-period or

**Table 3. SEM path associations.**

*SEM path associations: History of adolescent pregnancy → Adulthood health and parenting practices*

| | Maternal Physical Health | Maternal Mental Health | Discourage of Negative Emotion | Harsh Parenting | Conflict Parent-Child Relationship |
|---|---|---|---|---|---|
| | B (SE) | B (SE) | B (SE) | B (SE) | B (SE) |
| Adolescent pregnancy | -.16 (.05)*** | .10 (.05)* | .02 (.05) | .01 (.05) | .04 (.05) |

*SEM path associations: Adulthood health and parenting context→ Child health outcomes*

| | Child Health | Internalizing Problem | Externalizing Problem |
|---|---|---|---|
| Maternal Physical Health | .19 (.05)*** | .02 (.05) | .01 (.05) |
| Maternal Mental Health | -.07 (.05) | .20 (.05)*** | .04 (.05) |
| Discourage of Negative Emotion | .001 (.05) | .13 (.05)** | .11 (.05)* |
| Harsh Parenting | -.14 (.05)** | .18 (.05)*** | .30 (.04)*** |
| Conflict Parent-Child Relationship | -.06 (.05) | .10 (.05)* | .19 (.05)*** |

*SEM Indirect Effect: History of Adolescent Pregnancy → Parent Contexts → Child Outcome*

| | Child Health | Internalizing Problem | Externalizing Problem |
|---|---|---|---|
| Ado Pregnancy → Maternal Physical Health → | -.031 (.013)* | -.003 (.008) | -.002 (.008) |
| Ado Pregnancy → Maternal Mental Health → | -.008 (.007) | .020 (.011)+ | .004 (.006) |
| Ado Pregnancy → Discourage of Negative Emotion → | .000 (.001) | .002 (.007) | .002 (.006) |
| Ado Pregnancy → Harsh Parenting → | -.001 (.01) | .001 (.009) | .002 (.015) |
| Ado Pregnancy → Conflict Parent-Child Relationship → | -.003 (.004) | .004 (.005) | .009 (.010) |

***Note.*** Ado Pregnancy = History of Adolescent Pregnancy.

+p < .10

*p < .05

**p < .01

***p < .001

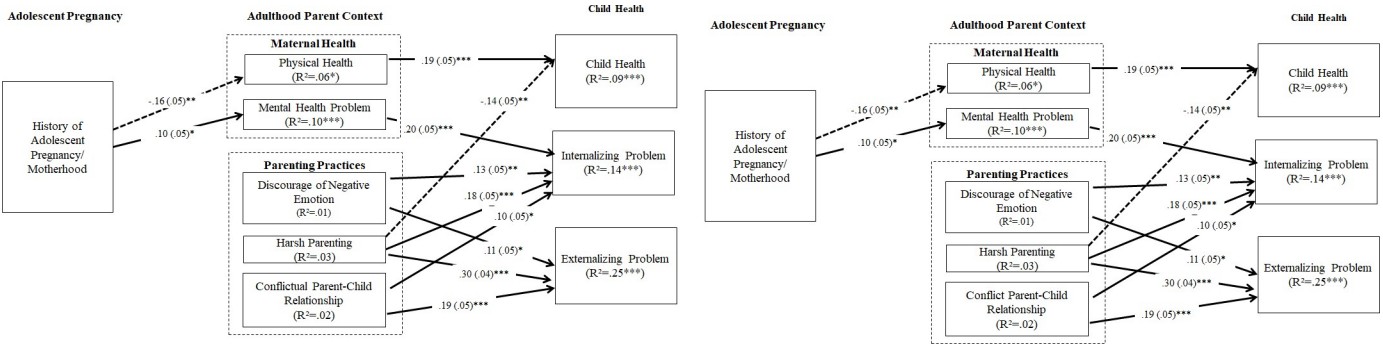

**Fig 2. SEM for association among adolescent pregnancy, maternal health, parenting, and child health.** *Note.* SEM model was applied for mediation indirect paths testing. SEM controlled for parent age, education, child age, food insecurity, number of children at home, and household size. Marital, employment, and child sex were not controlled because they were not associated with study variables. Standardized path coefficients and standard errors (in parentheses) are presented. A dash path indicates a negative association. Paths that did not reach significance are shown in Table 3. *p < .05 **p < .01 ***p < .001.

any main effects (adolescent motherhood and age-group) for the Harsh Parenting and Parent-Child Conflict measures.

**Offspring's health comparisons between two groups.**   On *offspring's health*, we found that offspring of adolescent mothers had significant higher internalizing problems than the offspring of non-adolescent mothers after adjusting for the confounders (see Table 2)). There was no group difference on child physical health or externalizing problems.

**A path model testing: The stress-SDH process for health disparities framework.**   To test the integrated path model (examining whether adolescent pregnancy is associated with poor subsequent adulthood health and parenting, which further influences child health/development outcomes), adjusted SEM was carried out. Fig 2 presents the standardized path coefficients for the significant paths and the $R^2$ values for each endogenous variable (maternal health, parenting, and child functioning). Overall, we found support of the path model. Global fit indices of the SEM model pointed to good model fit ($\chi2$ (3) = 4.82, p = .04; $\chi2$/df ratio = 1.60; CFI = .99, RMSEA = .040; 95% CI .00, .10). Furthermore, focused fit indices (standardized residuals and modification indices) revealed no theoretically meaningful points of stress on the model. Parental wellbeing and parenting constructs explained 9–25% of the variance in child physical and mental health outcomes. Findings indicated that heightened maternal history of adolescent pregnancy is significantly associated with poor maternal physical and mental health. Poor parental wellbeing was associated with increased internalizing and health problems of the children. Somewhat unexpectedly, after controlling for demographic and maternal health (physical and mental health), we found the history of adolescent pregnancy was not associated with parenting practice. However, consistent with parenting literature, all parenting variables were associated with child health outcomes (i.e., all associated with higher internalizing and externalizing problems). Harsh parenting was also associated with poor child health. SEM indirect effect analysis shown significant indirect effects for maternal adolescent pregnancy⟶ maternal physical health⟶ child physical health, E(SE) = -.03 (.01), *p* = .015. A trend significant indirect effect for maternal adolescent pregnancy ⟶ maternal mental health ⟶ child internalizing problems, E(SE) = .02 (.01), *p* = .065.

For parenting moderation, we found no support for the mediation mechanism. Although adolescent pregnancy has an influence on parents' emotional socialization (i.e., discourage negative emotion), it was not directly associated with child outcomes. However, consistent with parenting literature, we found support that harsh parenting and conflicted parent-child relationships were associated with poor child mental health.

## Discussion

This study contributes to the existing literature in three ways. It examines longer term impacts of adolescent motherhood history on women's adulthood health and parenting, as well as their children's health in a SSA country. This study also contributes to an understanding of two mechanism questions. We study the age-period effect by examining whether cumulated stressors and negative SDH experiences in adolescent mothers lead to deterioration of health and parenting as women become older. We also test an Integrated Stress-SDH Process for Disparities framework and examine whether early adolescent motherhood history set the path for poor adulthood health and parenting, which further lead to health disparities in offspring.

Adolescence is a difficult period in one's development [21] and is also a period when many risk or protective behaviors are consolidated [43]. Due to enormous structural and health disparities, not socio-cultural contexts provide optimal opportunities to channelize adolescent capabilities in the right direction [32, 44]. UN figures estimate over 16 million girls worldwide give birth between ages 15–19 and around 1 million before the age of 15 in LMICs, representing 20–30% of adolescent population) [45, 46]. Our community sample showed 32% of women with history of adolescent pregnancy/motherhood, which was similar to the prevalence reported in the literature [45]. Adolescent pregnancy and motherhood and associated health risks diminish peak fitness and lifetime health [32]. Adolescent motherhood, in particular, leads to early adultification and when 'children start having children', parenting is often compromised and of least relevance due to other compelling problems.

### Adolescent mothers have poor maternal mental health as adults

Our sample clearly shows that women with a history of adolescent motherhood have poor physical and mental health outcomes as adults. All three frameworks, the family stress model, social determinants of health, and adverse childhood experiences, that we have discussed here provide critical rejoinders to why adolescent pregnancy and motherhood are extraordinarily disabling life events. As adolescent motherhood brings along with itself several interconnected challenges (psychological, social, familial, and health), their abilities are compromised from the start to focus on the children and their own development. Social isolation and stigma are critical issues for adolescent mothers globally and this experience is more exaggerated in the informal settlements of Kenya where food insecurity, poverty, crime, violence, and abuse are rampant. These adversities reduce adolescent mothers' opportunities to seek life-skills, education, and livelihood support. It has been demonstrated that adolescents who become pregnant have internalizing tendencies such as low self-esteem, depressive symptoms, and anxiety. Social support, positive partner relationships, and education and employment might become protective factors for the adolescent mothers' offspring.

### Long term solutions towards poverty eradication, access to care, and gender equity

While breaking down social determinants of health into multiple categories is helpful in framing multilevel issues, ultimately, the focus needs to be towards more sustainable poverty eradication measures [47] in countries where adolescents do not have access to food, health care or education. Systems and services research have to pave the way for more focused work on addressing hunger and adolescent girls (including the girl child's) right to timely and quality receiving sexual and reproductive health services. A number of development frameworks have mapped now cumulative chronic poverty increases the odds for continued adverse experiences [48]. Health and gender equity for Sub-Saharan African context also implies strengthening the

State's functioning and allowing it to take charge of the overall human rights of its citizens [43].

## Cumulated stress-process and WHO SDH theoretical frameworks addressing challenges of early adolescent motherhood and its later impact

Both Stress-Process and WHO SDH framework contextualize health as a social phenomenon allowing for a discussion on universal health coverage and equity issues [48–50]. Collective social action and distribution of social and political power in the service of reducing health disparities and cumulated stressors is critical for deterioration of health outcome [51]. This study found some support for this hypothesis. We found an age-period effect on women's mental health on one parenting outcome (Discourage of Negative Emotion Expression in parenting). Specifically, Kenyan mothers' mental health tends to be deteriorating as they got older (due to cumulated adversity/stress in living in low-resource poor SDH contexts), and women with history of adolescent pregnancy have even worse mental health outcomes during adulthood than women without history of adolescent pregnancy. For *parenting*, we found a significant age-by-adolescent motherhood group interaction effect, which suggests a potential deterioration of parenting as worm aging in women with history of adolescent pregnancy.

## Path model testing

In the mechanism testing, the SEM model provides some support for the Integrated Stress-SDH Process for Health Disparities. We found the history of adolescent motherhood is significantly associated with poor adulthood physical and mental health, which also contributes to poor child physical health and internalizing problem. We did not find any significant links between the history of adolescent motherhood and parenting, which might be due to high correlations between maternal health and parenting indicators (as indicated in our SEM findings). After excluding the maternal health effects, there is little variance remain to explain the link between history of adolescent pregnancy and parenting. This finding suggests the importance of prioritizing maternal physical and mental health in reducing negative consequences on offspring. Although we did not find a significant link between the history of adolescent pregnancy and parenting, we do found a significant association between parenting and child health outcomes in expected ways. Consistent we the parenting literature, we found three areas of negative parenting are significantly associated with poor child health and more internalizing problems. Findings support the cross-cultural consistency in the theoretical models that underline women and child health disparities.

## Implications for improving maternal mental health interventions in Sub-Saharan Africa

There are several implications we can draw from this study despite its obvious limitations of being a cross-sectional design. In our current methodology, there is no implied causality that can be imputed between adolescent motherhood and poor maternal wellbeing and child health outcomes. Despite this drawback, we know that motherhood in adolescence is an adverse event as it takes away an opportunity for further development and growth for the adolescent girl. Instead, she is pushed into a life of adversity and high deprivation. This impacts both the adolescent and her child. Efforts have to be made to ensure that young girls know how to protect themselves and if they become pregnant, sound support and care is available to them. Avoiding a second early pregnancy is known to be a critical protective factor. The social determinants framework allows us to see that adolescent motherhood is associated with poor

parenting and child health outcomes. Adolescent mothers need greater parenting support. Early childhood development programs can integrate exposure to parenting for such vulnerable mothers. The current widespread application of Universal Health Coverage might also mean that psychosocial support interventions for adolescent mothers can be offered within ANC and MNCH clinics in primary health care settings. Another such recommendation of relevance to SSA is implementing program addressing the SDH through the development of policies enhancing health equity, gender equity, human rights and poverty reduction [51]. This itself would target many disadvantaged women, including those who became mothers during their adolescence.

## Supporting information

**S1 Study.**
(SAV)

## Author Contributions

**Conceptualization:** Manasi Kumar.

**Formal analysis:** Keng Yen Huang.

**Funding acquisition:** Manasi Kumar, Keng Yen Huang.

**Methodology:** Manasi Kumar, Keng Yen Huang.

**Writing – original draft:** Manasi Kumar, Keng Yen Huang.

**Writing – review & editing:** Manasi Kumar, Keng Yen Huang.

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
