## [Decision Letter · Decision Letter 0]

14 Oct 2020

PONE-D-20-09731

Impact of being an Adolescent Mother on Subsequent Maternal Health, Parenting, and Child Development in Kenyan low-income and high adversity informal settlement context

PLOS ONE

Dear Dr. Kumar,

Thank you for submitting your manuscript to PLOS ONE. After careful consideration, we feel that it has merit but does not fully meet PLOS ONE’s publication criteria as it currently stands. Therefore, we invite you to submit a revised version of the manuscript that addresses the points raised during the review process.

Your paper has been reviewed carefully by two peer reviewers with expertise in your field.  Please attend carefully to their suggestions and comments, particularly those of Reviewer 1, which are more extensive.  In addition, as editor I would raise the issues that the clarity of the study's objectives would be enhanced if outcomes for mothers and children were described separately.  Adolescent pregnancy is the exposure variable, but it is treated as a maternal outcome variable in the Measures section.  The Sample is not representative, respondents were not chosen on a probability basis.   The SEM models claims a mediating effect on child physical health, but since there was no main effect, this does not meet criteria (Barron and Kenney).  Figure 1 a and b are titled "Paternal" but I believe this is a typo, please correct.  Finally, the manuscript must be edited for scientific English grammar; there are problems with subject-verb agreement, sentence structure, articles (lack of), etc.

We look forward to receiving your revised manuscript.

Kind regards,

Ellen L. Idler

Academic Editor

PLOS ONE

Journal Requirements:

2. Please upload a copy of Figure 3, to which you refer in your text. If the figure is no longer to be included as part of the submission please remove all reference to it within the text.

Reviewers' comments:

Reviewer's Responses to Questions

**Comments to the Author**

1. Is the manuscript technically sound, and do the data support the conclusions?

Reviewer #1: Partly

Reviewer #2: Yes

2. Has the statistical analysis been performed appropriately and rigorously? 

Reviewer #1: Yes

Reviewer #2: Yes

3. Have the authors made all data underlying the findings in their manuscript fully available?

Reviewer #1: Yes

Reviewer #2: Yes

4. Is the manuscript presented in an intelligible fashion and written in standard English?

Reviewer #1: No

Reviewer #2: Yes

5. Review Comments to the Author

Reviewer #1: I applaud the authors’ work on this important topic of long-term impacts of adolescent pregnancy on a range of health and development outcomes.

However, my overall impression of this manuscript is that – while the methodology is generally sound – the study rationale, theoretical framing, and interpretation and discussion of results are weak. Below please find general comments and specific questions.

The conceptual framework is unclear. I do not see Figure 1, as referenced by the authors as illustration of how they conceptualize SDOH in this context. Further, structural determinants (e.g. education) appear to be used as control variables, rather than covariates of substantive interest. This is not consistent with a SDOH logic, which would center the measurement and effects of macro-level features of the local society – which appears to be quite unique (see below). I also note that the authors’ later mention in the discussion “All three frameworks, family stress model, social determinants of health and adverse child experiences, that we have discussed here provide critical rejoinders….” I do not see any reference to the family stress model or adverse child experiences, as they relate to the study.

In the background section, the authors conflate the literature on early/adolescent motherhood, unintended motherhood and single motherhood – yet these are distinct, although sometimes overlapping, phenomena. I encourage the authors to be more specific and explicit in their summary of the existing literature, particularly from LMICs. In particular, I would want to know how this manuscript advances our knowledge of the long-term effects of adolescent motherhood in the context of Kenya/East Africa.

In many places, the discussion of results is extremely general and not specific to the study itself. For example, the authors suggest that the ultimate focus needs to be on sustainably poverty eradication measures. This statement is not clearly supported by the study design or results. What measures of poverty are included? The second section on WHO theoretical frameworks reads as a generic summary of existing frameworks, rather than a specific discussion of how the study results advance discussion of development priorities. The authors mention a Figure 3, which I do not see included in the manuscript.

Finally, I am curious about the study site. This appears to be a unique setting. Indeed, the authors explicitly mention that this is a “low income, high adversity informal settlement context” in the manuscript title, and of course, they reference structural and social conditions that influence health in their discussion of the SDOH framework. Yet, there is inadequate explanation of the context, and no apparent measurement of local social or structural conditions. For example, if this was an informal settlement, why were no measures of migration included? The authors mention it is a high-adversity context. Why are no measures of violence or adversity included? What is relevant about the study site? I would want to know why was the study conducted here, how did measures account for the local setting, and how might the results of this study be affected by the socio-cultural conditions of place.

General questions and comments

• The authors’ note that the Kenya government recorded 350,000 adolescent pregnancies between 2016-2017 and that this is a rise from previous estimates. Yet the previous estimate was published in 2017? How do we know this is a substantive increase, and not just general fluctuation? Is there available trend data that the authors could use to make this point better?

• Among women who were adolescent mothers, were the children sampled the children who were born before the mother was age 18?

• The authors describe this as a female caregiver sample. Please provide greater detail. Are they caregivers because they were bringing children to the health clinic?

• Were interviews conducted by women?

• Please include measurement details for all covariates. For example, how was food insecurity measured?

• Use of causal language should be avoided, given the cross-sectional data. For example, the regression models do not provide evidence of impact of adolescent motherhood on maternal education, employment and food security. They provide evidence for association. The pathway could be bidirectional.

• The study appears to use measures of child’s biological sex, not gender. I encourage the authors to make edits where appropriate.

• Why are figures of ANCOVA analyses presented only for maternal health, and not also for parenting practices and child health? Further, the results also seem to indicate need for mental health interventions with older women, not only preventative work with younger women.

• Why is food insecurity not included as a control variable for regression and SEM models?

• In many places, there is a disconnect between sequential sentences, which inhibits flow of reading and comprehension. For example, “Furthermore, adolescent pregnancy also has severe medical and obstetric complications. One study from UK [sic] found that the early age at first birth is associated with poorer mental health…” In this example, I would be interested to learn also about the medical and obstetric complications and be pointed toward key references.

• What do you mean by references to conflict marriage? Abusive relationships? What is the prevalence of IPV in this sample? If it was measured, I would strongly recommend including IPV in the analysis, given its potential role as a confounding variable.

• The manuscript requires a comprehensive copy-edit. There are extensive grammatical and spelling errors throughout the document.

Reviewer #2: Thank you for the opportunity to review this very interesting review study. The manuscript was well-organized. There are a number of aspects which require revision to make this paper suitable for publication

1. Sampling does not clear, what the sampling method and the sample size calculation

2. why the sample size of the two groups is different?

3. Inclusion criteria should be clear, the participants were mother who over 18 years or younger when they giving birth? Please clearly define of adolescent mother

4. Give more detail related study setting located

5. Explain the aims at the beginning of discussion part

6. The discussion of findings is good, but rather need strengthen discussion related result especially for child internalizing problems and child physical health

7. Discussion related Adolescent mothers have poor maternal mental health as adults only focus on results, need more strengthen

6. PLOS authors have the option to publish the peer review history of their article (what does this mean?). If published, this will include your full peer review and any attached files.

Reviewer #1: No

Reviewer #2: No

---

## [Author Response · Author response to Decision Letter 0]

11 Dec 2020

Dear Editor we have enclosed a document with files that has point by point response to both reviewers feedback and comments 

regards manasi

---

## [Decision Letter · Decision Letter 1]

4 Jan 2021

PONE-D-20-09731R1

Impact of being an Adolescent Mother on Subsequent Maternal Health, Parenting, and Child Development in Kenyan low-income and high adversity informal settlement context

PLOS ONE

Dear Dr. Kumar,

Thank you for submitting your manuscript to PLOS ONE. After careful consideration, we feel that it has merit but does not fully meet PLOS ONE’s publication criteria as it currently stands. Therefore, we invite you to submit a revised version of the manuscript that addresses the points raised during the review process.

Both reviewers agree that the revision has greatly improved the paper.  Please see the attached comments by Reviewer 1, which request that you provide some more background on the theoretical frameworks that structure the study.  Also, this review requests that you include the tested, but nonsignificant pathways in your SEM model, and review the manuscript carefully for English grammar and sentence structure (specific issues are noted).  

We look forward to receiving your revised manuscript.

Kind regards,

Ellen L. Idler

Academic Editor

PLOS ONE

Reviewers' comments:

Reviewer's Responses to Questions

**Comments to the Author**

1. If the authors have adequately addressed your comments raised in a previous round of review and you feel that this manuscript is now acceptable for publication, you may indicate that here to bypass the “Comments to the Author” section, enter your conflict of interest statement in the “Confidential to Editor” section, and submit your "Accept" recommendation.

Reviewer #1: (No Response)

Reviewer #2: All comments have been addressed

2. Is the manuscript technically sound, and do the data support the conclusions?

Reviewer #1: Yes

Reviewer #2: Yes

3. Has the statistical analysis been performed appropriately and rigorously? 

Reviewer #1: Yes

Reviewer #2: Yes

4. Have the authors made all data underlying the findings in their manuscript fully available?

Reviewer #1: Yes

Reviewer #2: Yes

5. Is the manuscript presented in an intelligible fashion and written in standard English?

Reviewer #1: No

Reviewer #2: Yes

6. Review Comments to the Author

Reviewer #1: Thank you for your detailed attention to reviewer comments. Below please find outstanding questions and recommendations:

What is the Integrated Stress –SDH Process to Health Disparities Framework? The authors do not provide clear explanation for readers who may be unfamiliar with these guiding theories. Suggest including a section in the background that outlines the framework itself (perhaps before you provide review of the literature).

The sample is not representative, and should not be described as such (pg. 6)

Table 2 – child overall health mean for total sample appears to be a typo?

The authors might consider including a third table with all direct and indirect effects, including those that are non-significant, to provide readers with a comprehensive set of results from final path models.

In some places, the sentence structure and grammatical errors inhibit comprehension of the authors’ arguments (e.g. p. 10 “due to enormous structural and health disparities, not socio-cultural contexts…etc” – not clear what is meant here).

My earlier comment about “all three frameworks” has not been addressed (p. 11). Family stress model, and adverse childhood experiences are discrete theoretical frameworks which are alluded to but not explained in great detail. I would recommend dropping this sentence from the discussion.

Reviewer #2: Thank you for the opportunity to review this very interesting review study. The manuscript was well-organized and I recommendation to accept this manuscript

7. PLOS authors have the option to publish the peer review history of their article (what does this mean?). If published, this will include your full peer review and any attached files.

Reviewer #1: No

Reviewer #2: No

---

## [Author Response · Author response to Decision Letter 1]

3 Mar 2021

Reviewer #1: Thank you for your detailed attention to reviewer comments. Below please find outstanding questions and recommendations:

• What is the Integrated Stress –SDH Process to Health Disparities Framework? The authors do not provide clear explanation for readers who may be unfamiliar with these guiding theories. Suggest including a section in the background that outlines the framework itself (perhaps before you provide review of the literature).

We now clarify that Family Stress Model and SDH as two different frameworks. The Family Stress Model posits that level of stress influences parental health, quality of parenting, and child development) (McLoyd, et al., 1998; Conger et al., 2010), and the Social Determinants of Health (SDH) framework posits social and structural environment and available resources influence population health) (WHO, 2008, 2016). Both frameworks recognize complex interplay and stress experiences of the social and economic systems that people live in and their impacts on health of family members. We have revised the descriptions in the section. 

• The sample is not representative, and should not be described as such (pg. 6)

We now clarify the concern by delete the descriptions “Our sample is a representative community sample…….” and “…in this retrospective cohort study”.

• Table 2 – child overall health mean for total sample appears to be a typo?

We apologize for the typo. We have corrected the mean values. 

• The authors might consider including a third table with all direct and indirect effects, including those that are non-significant, to provide readers with a comprehensive set of results from final path models.

We now add Table 3 to show all indirect and direct effects.

• In some places, the sentence structure and grammatical errors inhibit comprehension of the authors’ arguments (e.g. p. 10 “due to enormous structural and health disparities, not socio-cultural contexts…etc” – not clear what is meant here).

We have edited the manuscript throughout and break down the long sentences to make them easier to read.

My earlier comment about “all three frameworks” has not been addressed (p. 11). Family stress model, and adverse childhood experiences are discrete theoretical frameworks which are alluded to but not explained in great detail. I would recommend dropping this sentence from the discussion.

Please see above. We revise the section and make the descriptions more clear.

---

## [Editor Report · Decision Letter 2]

8 Mar 2021

Impact of being an Adolescent Mother on Subsequent Maternal Health, Parenting, and Child Development in Kenyan low-income and high adversity informal settlement context

PONE-D-20-09731R2

Dear Dr. Kumar,

We’re pleased to inform you that your manuscript has been judged scientifically suitable for publication and will be formally accepted for publication once it meets all outstanding technical requirements.

Kind regards,

Ellen L. Idler

Academic Editor

PLOS ONE
---

## [Editor Report · Acceptance letter]

19 Mar 2021

PONE-D-20-09731R2 

Impact of being an Adolescent Mother on Subsequent Maternal Health, Parenting, and Child Development in Kenyan low-income and high adversity informal settlement context 

Dear Dr. Kumar:

I'm pleased to inform you that your manuscript has been deemed suitable for publication in PLOS ONE. Congratulations! Your manuscript is now with our production department. 

Kind regards, 

on behalf of

Professor Ellen L. Idler 

Academic Editor

PLOS ONE